# Thermal Annealing Effect on Elastoplastic Behaviour of Al/Cu Bimetal during Three-Point Bending

**DOI:** 10.3390/ma17184637

**Published:** 2024-09-21

**Authors:** Robert Uścinowicz

**Affiliations:** Department of Mechanics and Applied Computer Science, Faculty of Mechanical Engineering, Bialystok University of Technology, 45C Wiejska, 15-351 Bialystok, Poland; r.uscinowicz@pb.edu.pl; Tel.: +48-571443040

**Keywords:** Al/Cu bimetal, thermal annealing, three-point bending test, elastic constants, springback, elastoplastic deformation, micrography

## Abstract

This paper presents the results of experimental studies on the effects of temperature and time of annealing on the elastoplastic properties of bimetallic aluminium–copper sheets. Mechanical tests were carried out on flat samples previously heated to temperatures of 250, 350, 450, and 500 °C for 40, 90, and 150 min. At the beginning of the tests, the elastic constants and internal friction energy were determined after thermal exposure using the impulse vibration exposure method. Further tests were carried out on the same samples using the three-point bending test. Based on the tests, the following quantities were determined and analysed: elasticity angles, translocations of the neutral axes of the cross-sections of samples, and changes in the values of bending moments plasticizing the extreme layers of bimetallic Al/Cu samples resulting from thermal interactions. The final part of this paper presents the results of measurements of the thickness of diffusion zones at the interface and their effect on the stability of the joint after annealing. The studies that were conducted indicate the dominant influence of the thermal factor on the properties of the Al/Cu bimetal above the temperature of 350 °C, which leads to the weakening of its strength and the degradation of the structure at the metallic phase boundary.

## 1. Introduction

With the development of new technologies, the demand for new, structurally complex materials with special pre-designed physical properties increases. Many of these features are present in layered composite materials produced based on various metals and metal alloys by permanently combining them. The inseparability of the layer connection is one of the important features determining the suitability of a layered metal composite for technical and non-technical applications. Its ease of processing, including subsequent technological plastic shaping, is also considered. In addition to the proper selection of the component metals for the composite and the technology for their connection, it is necessary to test the composite in operating conditions, as well as extremely unfavourable ones, including a comprehensive assessment of the reactions occurring at the interface of the layer connection.

One of the most frequently used layered composites is aluminium–copper bimetal. It has several advantages, the most important of which are high electrical and thermal conductivity, high corrosion resistance, low density, and relatively low cost of production by cold or hot rolling. The use of Al/Cu bimetallic elements in electrical engineering and electronics allows for weight reductions of 40–50% without significantly reducing the ability to conduct electricity. At the same time, it is possible to save 30–50% of the cost compared to monolithic copper [1]. For these reasons, Al/Cu bimetal is often used in yoke coils in TVs, armoured cables, heat sinks, cooling fins, heat exchangers, and busbar connectors.

The author’s observations indicate that the use of Al/Cu bimetal in electrical engineering and heating is also associated with the risk of unforeseen temporary and sometimes permanent thermal overloads of structural elements, which causes irreversible and difficult-to-predict effects on its structure, especially at the interface between layers [2,3,4]. In addition, it is associated with the formation of brittle intermetallic compounds, which worsens the mechanical and electrical reliability of the Al/Cu composite. Studying the interphase structure and properties of Al/Cu clad composites after thermal exposure has been the aim of several works [1,5,6]. Nevertheless, the assessment of the effect of interphase and intermetallic compounds on the mechanical properties of Al/Cu composites, especially those produced by cold and hot rolling, is not complete and is still relevant.

Structural elements made of bimetals are very often subjected to elastic–plastic bending in operating conditions, resulting from both mechanical and thermomechanical loads that cause complex stress states in their structure, making the interpretation of strength very difficult. The issue of bending bars made of laminated metals has been the subject of interest for many researchers. In the work of Arslan and Sulu [7], the problem of elastoplastic pure bending of a two-layer bar was solved. Cases of initial plasticization of a bar consisting of various combinations of component metals (steel/aluminium) with variable geometry were considered there. It was found that the location of the plasticized layer may appear in various places in the composite rod. Equally interesting is the earlier work of Verguts and Sowerby [8] on the theoretical and practical case of the pure bending of glued sheets of two different metals under the conditions of a plane deformation state, considering the Bauschinger effect in the analyses. The more difficult problem of three-point bending of an expanded–tapered sandwich beam was considered analytically and numerically in the work by Magnucki et al. [9], in which an analytical model was developed based on the broken-line hypothesis. Another work [10] by this author was also devoted to the analytical and numerical solutions of the problem of bending of two-layer T-beams under continuous load. An interesting analytical model based on numerical analysis can be found in the work of Li et al. [11], where the properties of the 2205/AH36 BC bimetal under three-point bending were investigated considering many factors, including the interfacial diffusion zone.

Bimetals based on aluminium and copper as model materials under three-point bending conditions were the subject of research by Lee and Lee Kwon [12]. In turn, in the Al/Cu/Al three-layer composite, bending effects were presented in the works of Li et al. [13] and Kim and Hong [1,14,15]. The case of bending the Al/Cu bimetal in the “V”-type punch–die bending process was described in the work of R. Srinivasan and Karthik Raja [16] and the “U”-type die–punch in the work of Gautam et al. [17].

Examining the effect of both short-term and prolonged thermal impacts on the layered aluminium–copper composite, which performs specific functions in technology, is an important issue from both the technical and scientific points of view. The complexity of its structure, built of two metals and a brittle (adhesive) interface, implies the difficulty of the problem, which will be considered below experimentally under the conditions of a three-point bending test. The aim of the research conducted and described in this paper was to qualitatively and quantitatively assess the effects of the temperature and time of annealing on the elastic–plastic deformation ability of the aluminium–copper bimetal. The paper considers such issues as the springback effect of the Al/Cu bimetal after exposure to heat, the translocation of the neutral axis as a result of annealing, the identification of changes in the plasticizing moment values of the extreme layers of the bimetal, and the assessment of the growth scale of the diffusion layer at the phase separation boundary.

## 2. Materials, Test Specimens, and Testing Procedures

All tests were carried out on samples cut from aluminium–copper (Al/Cu) sheets in the direction of rolling, and their final shape (cuboid) was obtained by milling. An Al/Cu sheet with nominal dimensions of 400 × 1000 mm and a thickness of approximately 4 mm was produced by combining a sheet of electrolytic copper (CW004A) with a sheet of pure aluminium (AW-1050A) by hot rolling. The nominal thickness of the aluminium layer in the Al/Cu bimetal was 1.95 mm. The chemical compositions of the sheets are presented in Table 1 and Table 2. The thickness of individual layers in bimetallic samples was measured using graphical analysis methods, recording their microscopic images. The average percentages of the components in the Al/Cu bimetal in the delivered state (unprocessed) were approximately 51% Cu and 49% Al. Figure 1 shows the geometric dimensions of the Al/Cu sample, an SEM (Scanning Electron Microscope) photograph of the interface, and the linear distribution of Al and Cu in the initial state, i.e., without thermal interactions implemented in the experiment. The samples described above were divided into two groups. The first group consisted of samples that were tested after heating at temperatures of 250, 350, 450, and 500 °C for 40, 90, and 150 min. The second group consisted of samples that were not subjected to the heating process (marked in this paper as 22 °C).

The experimental research included three stages, which were carried out on the same Al/Cu bimetallic samples described above (Figure 1a):

Stage I—The determination of the elastic properties and internal friction parameters of Al/Cu using the impulse excitation technique;

Stage II—Three-point bending tests on Al/Cu samples;

Stage III—Microscopic analysis of the layer connection zone before and after the plastic deformation of Al/Cu samples.

## 3. Elastic Properties of Al/Cu Bimetal after Thermal Exposure

A non-destructive method was used to assess the elastic properties of the Al/Cu bimetal, which involves the analysis of resonance frequencies as an effect of a mechanical impulse. Thanks to this method, Young’s (*E*) and shear (*G*) moduli, the internal friction parameter (*Q*^−1^), and the damping coefficient were determined. The principles of this method are described in the ASTM E1876–22 standard [18]. The tests were carried out at room temperature using the RFDA (Resonant Frequency and Damping Analyser) measurement set from IMCE [19]. The basis of this method is the measurement of the fundamental resonance frequency of the acoustic signal triggered by a single mechanical excitation caused by an impulse tool hitting the sample. During the test, the sample with the geometry and shape shown in Figure 1a lay on supports. It was simultaneously three-point bent and twisted in the direction perpendicular to the lamination. At the same time, a transducer in the form of a microphone detected and recorded the acoustic signal resulting from the mechanical (elastic) vibrations of the sample and transformed them into electrical signals saved in digital form in a computer. The algorithms of the computer program, using the Fourier transform, analysed the spectrum of resonant frequencies and determined the elasticity coefficients, loss coefficients, and damping coefficients for the identified signal frequencies. This method is also applicable to composites and anisotropic (orthotropic) and inhomogeneous materials after considering their morphology, e.g., the distribution of layer orientations and the durability of interfacial bonds, as described by Uścinowicz [20] and Song et al. [21]. Additionally, the method allows for the identification of the integrity of the medium, providing information about its elasticity in both qualitative and quantitative terms (Roebben et al. [22,23]).

For cuboid-shaped samples (Figure 1a) and signal measurements in simultaneous bending and torsion mode, the values of Young’s (*E*) and shear (Kirchhoff) (*G*) moduli were calculated from the following relationships [19]:*E* = 0.9465 (*m f*_b_ ^2^/b) × (*L*^3^/*t*_o_^3^) H, *G* = (4*L m f*_t_/*b t_o_*) × [B/1 + A],(1)
where *m* is the mass of the sample; *L*, *t_o_*, and *b* are the length, thickness, and width of the rod, respectively; *f*_b_ and *f*_t_ are basic resonance frequencies during bending and torsion of the sample, respectively; and H, A, and B are calculation correction factors.

Additionally, the basic resonance frequencies, *f*_b_ and *f*_t_, damping coefficients (internal friction parameter *Q*^−1^), and losses were calculated.

In assessing the impact of thermal effects and their duration, the internal friction parameter *Q*^−1^ was helpful, defined by the following relationship:*Q^−^*^1^ = Δ*W*/2π*W*,(2)
where *W* and Δ*W*, respectively, are the energy accumulated and lost per unit volume of the vibrating body during one period.

Fourier analysis was used to calculate *Q*^−1^, which was calculated using the formula
*Q^−^*^1^ = *k*/π *f*_b(t)_,(3)
where *k* is an equation parameter.

The values of the resonance frequency *f*_b(t)_ that corresponded to vibrations were also determined with the equation
x(*t*) = Ae^−*k*t^ sin (2π *f*_b(t)_ *t* + *φ*),(4)
where *A* and *φ* are parameters of Equation (4), *f*_b(t)_ is the resonance frequency in the three-point bending and torsion mode, and *t* is the time parameter.

The loss of the elastic properties of the Al/Cu bimetal because of the influence of the temperature and annealing time can be seen in Figure 2a and Figure 3a, which show the change in the value of Young’s modulus (*E*). For the heating time *t* = 40 min. in the temperature range of 250–450 °C (Figure 2a), there was an almost linear decrease in the Young’s modulus value from 100 GPa to 90.5 GPa. In contrast, the increase in annealing temperature to 500 °C (in this case) does not cause significant changes in the *E* value. Samples annealed for 90 and 150 min. initially show a clear decrease in value in the temperature range of 250–350 °C from approximately 97 GPa to 90 GPa. Above 350 °C, the value sits at a level of around *E* = 91 GPa. It should be emphasized that in the entire range of 250–500 °C, the characteristics of changes in the *E* value for *t* = 90 and 150 min. are almost identical. The annealing time has a significant influence on the E values only for temperatures of 250 and 350 °C at *t* = 40 min, because then these values of *E* decrease by 2.6 GPa and 4.3 GPa, respectively (Figure 3a). The value of Young’s modulus for time *t* = 40 min. and temperature *T* = 250 °C is practically the same as for unannealed samples (*T* = 22 °C).

In the case of the shear modulus (*G*), the differences in its value caused by the increase in temperature are small and range from approx. 32 to 34 GPa, and, unlike Young’s modulus, the *G* value increases with the increase in temperature (Figure 2b). In the temperature range of 450–500 °C, the values of the *G* modulus for the tested heating times decrease and are similar to one another. In the case of *t* = 90 and 150 min., the characteristics describing changes in the shear modulus as a function of temperature have a similar shape (Figure 2b). The values of the shear modulus after annealing at temperatures of 250 and 350 °C for *t* = 40 min. are the same (32.7 GPa) and slightly different from the values for unannealed samples (Figure 3b). The greatest effect of the annealing time on the shear modulus value was noted for annealing times of 90 and 150 min. and was in the temperature range of 250–450 °C.

A valuable parameter enabling the determination of changes occurring in the structure of the Al/Cu bimetal is the internal friction coefficient *Q*^−1^. It allows one to capture qualitative changes that result from the impact of heat on a two-layer metal composite. Figure 4a shows the changes in the *Q*^−1^ value with increasing annealing temperature. The change in the *Q*^−1^ parameter with increasing temperature is characterized by nonlinearity. This fact is confirmed in the work of Puscar [24], who observed this effect after exceeding a temperature of 0.5–0.6 *T*_m_ (*T*_m_—melting point), which occurs in the case of aluminium.

It was observed that for the three tested heating times, *t* = 40, 90, and 150 min., the internal friction value increased from an average level of approximately *Q*^−1^ = 1.7 × 10^−4^ at a temperature of 250 °C to values of 8.9 × 10^−4^, 11.4 × 10^–4^, and 10.8 × 10^−4^ at 500 °C, respectively (Figure 4a). It should be emphasized that the values of the *Q*^−1^ parameter for the three tested heating times at 250 °C did not practically differ and were slightly lower than in the case of the unheated bimetal (*Q*^−1^ = 2.4 × 10^−4^). Moreover, at a temperature of 250 °C, no diffusion process was observed at the aluminium/copper interface. For the remaining temperatures, *T* = 350, 450, and 500 °C, the *Q*^−1^ values were significantly higher, and the maximum occurred at *t* = 90 min, which was related to the diffusion process accompanying the formation of new phases with different morphologies. The highest value of the *Q*^−1^ parameter was obtained at a temperature of 500 °C and was 11.4 × 10^−4^.

## 4. Characterization of the Elastoplastic Properties of Al/Cu in the Three-Point Bending Test

The second stage of this research included a three-point bending test, which was carried out to the maximum deflection value *d*_max_ = 6 mm. The tests were carried out on the MTS Mini Bionix 858 hydraulic universal machine according to the recommendations contained in the technical standard ASTM E–290–22 [25]. The samples were loaded centrally, halfway between the supports and perpendicular to the lamination, using a special device. The distance between the supports was 60 mm. The force was applied to the copper layer. The movement speed of the actuator with a rounded tip was constant and amounted to 2 × 10^−2^ mm/s. These tests were carried out on the same samples that were tested using a resonance frequency analyser (non-destructive method). The measurements of deformations and displacements (deflections) of the measured parts of the samples were carried out using the Aramis digital image correlation system from GOM. The effectiveness of this strain measurement method was confirmed in the work of Govindasamy and Jain [26], who analysed the deformation processes of samples made of AA2024-T4 aluminium alloy strips under pure bending conditions. The deformation measurement was recorded on the side surface of the sample (along with its thickness).

From the tests, curves showing changes in the centrally applied force (*F*) causing the bending effect and the corresponding deflection (*d*) were obtained. Figure 5 shows exemplary three-point bending curves in the *F* − *d* system for samples heated for 90 min. at temperatures *T* = 250, 350, 450, and 500 °C and the unheated sample (*T* = 22 °C).

### 4.1. The Effect of Temperature on the Springback of Al/Cu Bimetal Samples

The distortion of the geometry or the desired shape of the Al/Cu bimetal resulting from the impact of heat is considered a manufacturing and operational defect. This unintentional change in the material geometry is identified, especially during bending. It may affect both the first and second assembly processes. One of the phenomena that is of great technological importance is springback. This results from the elastic properties of the metal as it moves back after the elastoplastic load is removed in order to recover some of the elastic energy. 

The springback effect is often discussed in the literature related to plastic processing, especially sheet metal stamping. It is believed that the springback of bimetals depends on the properties of the component metals, layer arrangement, interface properties and surface quality, and joining parameters and conditions [27], and this is an unintentional change in the geometry of the material, especially when bending. The effect of springback during bending in relation to an aluminium/stainless steel bimetallic sheet was described in the work of Yilam et al. [28], where the influence of the locations of the weaker and stronger bimetal layers relative to the punch on the springback angle, thickness, and radius of the sample during bending was considered. Similar considerations were made by Nikhare [29], who considered the thickness factor of the two-layer material. Gautam et al. [17] performed experimental tests and numerical simulations of the springback effect of a three-layer clad sheet during U-shaped bending. Hino et al. [30] conducted tensile and bending experiments on bi-layer bimetals consisting of pure aluminium and ferritic stainless steel, showing that the springback of sheet metal laminates is greatly influenced by the strength difference between the component layers of the laminate and the relative positions of the layers and the layer thickness ratio. Similar studies assessing the springback effect in relation to metal sandwich composites were carried out by Oya et al. [31] and Shayan et al. [32].

The support scheme of the Al/Cu sample before loading with a centrally applied force in the 3-point bending process is shown in Figure 6a. The springback in the tested Al/Cu bimetal was measured and analysed by measuring the difference in angles Δ*Θ* during bending in two states of sample deformation. The first concerned the angle *Θ*_1_ created by the sample arms (Figure 6b) before unloading in relation to the unloaded sample, which corresponded to the maximum deflection of *d* = 6 mm of the sample. The second condition determined the angle *Θ*_2_, which was measured similarly to the first case but after removing the load from the sample. The results of these tests for the range of temperatures and annealing times are presented in Table 3 and Figure 7.

As the temperature increased from 250 to 450 °C, the values of the angles Δ*Θ* increased for the tested annealing times (Figure 7a). However, as the temperature increased to 500 °C, there was a decrease in Δ*Θ*, which can be attributed to the already significant diffusion zone at the interface, characterized by reduced elasticity. The increase in Δ*Θ* for the annealing time *t* = 40 min was linear, while for the remaining *t* times, it was strongly nonlinear. The maximum value of Δ*Θ* was obtained for a temperature of 450 °C and amounted to 3.24 deg. for *t* = 150 min., Δ*Θ* = 3.11 deg. for *t* = 90 min., and Δ*Θ* = 3.01 deg. for *t* = 40 min. The value of the springback angle for unannealed samples was 0.41 deg., and it was about 0.17 deg. smaller than for samples annealed at 250 °C for 40 min.

Figure 7b allows the reader to assess how the annealing time influenced the value of Δ*Θ*. The observations allow us to conclude that increasing the time from 90 to 150 min for samples annealed in the temperature range of 250–450 °C slightly influenced the value of the angle Δ*Θ*. In turn, extending the time from 40 min to 90 min is significant for samples annealed at temperatures of 500 °C and 350 °C.

The Δ*Θ* angle values (springback) obtained from the experiment for the Al/Cu bimetal were not large and amounted to 3.24 deg. Similar values of angles were recorded by Gautam et al. [17] in their experimental studies of the SS-Al-SS trimetal (3.58 deg.) and FEM (Finite Element Method) simulations (3.18 deg.) for the bending direction in the direction of rolling, as well as by Yilamu et al. [28], who, for bimetal steel stainless steel–aluminium (SS-Al.), obtained 2.2 deg. There was no significant increase in the thickness of the samples at the point of greatest deflection *d*.

### 4.2. Identifying the Position of the Neutral Axis

The location of the neutral plane in two-layer metal composites is of great technological and operational importance. The most advantageous is its location at the interface. The importance of the location of the neutral layer was demonstrated in work by Yoshida et al. [33], who proposed an aluminium–chrome bimetal as a remedy for high stress values in chrome sheets. More extensive analytical, numerical, and experimental studies on the stress distribution and the position of the neutral axis in the 2205/AH36 bimetal during three-point bending can be found in the work of Li et al. [34], as well as in the older work on theoretical considerations by Majlessi and Dadras [35]. Some technical solutions of two-layer laminates using metals and polymers introduce an additional third transition layer—a buffer layer that eliminates delamination stresses, causing zero interfacial stress [36].

For layered metals, including the Al/Cu bimetal, an important research goal is the issue of the translocation of the neutral layer during three-point elastoplastic bending. The location of the neutral layer was determined for the two load states of the samples. The first state was related to the search for the *h*_n1_ coordinate describing the position of the neutral axis in the sample cross-sections in the elastic range of both the Al and Cu layers. The next condition concerned the loading of the samples in the range of elastic–plastic deformations, which corresponded to the maximum deflection of *d* = 6 mm. In this state, the location of the neutral axis in the sample cross-section was identified and described by the *h*_n2_ coordinate.

The position of the neutral axis in the Al/Cu cross-section of the sample in the elastic range was determined using the analytical method for the composite cross-section. An analytical method was used to determine the position of the neutral axis in the elastic range of the sample in the Al/Cu composite cross-section. The procedure required information about the Young’s moduli of the bimetal components, which were determined experimentally. The method for determining the moduli and their values is presented in [3], in which the same Al/Cu sheet was tested. The values of these moduli were then obtained for copper *E*_Cu_ = 158.6 GPa and aluminium *E*_Cu_ = 59.5 GPa. Calculations were performed for the cross-sectional geometry of unannealed and annealed samples in the temperature range of 250–500 °C.

In the first loading condition of the samples, the position of the main central axis of inertia of the cross-section (in the geometric sense) was determined for each of them. The weight resulting from the ratios of Young’s moduli of aluminium and copper was determined and used to determine the weighted static moments *S*_zw_ and the weighted cross-sectional areas *A*_w_ relative to the main central axes. The average values of these quantities were *S*_zw_ = 24.5 mm^3^ and *A*_w_ = 55.4 mm^2^, respectively. In the next stage, the position of the weighted axis in relation to the central main system Z_c_ − Y_c_ was determined by first determining the distance Δ*z* between the Z_c_ and Z_n_ axes (Figure 8). The weighted moment of inertia about the Z_n_ axis, which was the neutral axis of the Al/Cu double-layer cross-section, was also calculated for each sample. The weighted average axial moment of the cross-section was *I*_w_ = 62.0 mm^4^. The position of the neutral axis was determined by the parameter relative to the extreme bottom Al layer (Figure 8). Knowledge of the value of the force loading the sample allowed for the calculation of the maximum bending moment corresponding to the proportionality limit in the elastic range. The obtained geometric and physical quantities were calculated independently for each tested sample and are presented in Table 4. Due to the different dimensions of the Al and Cu layers in the sample cross-sections, the *h*_n1_ parameter values were calculated individually for each sample. The dimensional selection carried out at the beginning of the tests assigned samples with similar geometric dimensions of Al and Cu layer cross-sections to the same group of temperature interactions. This action allowed for averaging the *h*_n1_ parameters. The average *h*_n1_ value for all samples was approximately 2.45 mm.

For samples annealed in the range of 250–500 °C, with the maximum sample deflection *d* = 6 mm, the position of the neutral axis marked with the h_n2_ coordinate was determined by analysing the distribution of elastoplastic strains in the direction of normal stresses using the DIC technique and the Correlation program from GOM. The determined *h*_n1_ and *h*_n2_ values are included in Table 4.

The Δ*h* parameter, defined as the difference between the values of *h*_n2_ and *h*_n1_, was used to analyse the translocation of the neutral axis. For a better assessment of the changes occurring in the positions of the neutral axes in the cross-sections of bimetallic samples as a result of the annealing temperature and time, Figure 9a,b were prepared. They allow for a more accurate assessment of changes in the position of the neutral axis as the load increases (for two load states) without considering differences in sample geometries.

Figure 9a shows that for unannealed samples and samples annealed at 250 °C, the Δ*h* values were in the range of 0.15–0.21 mm. The lowest value, Δ*h* = 0.15 mm at a temperature of 250 °C, was recorded for an annealing time of 150 min. In turn, the highest value of Δ*h* occurred during the annealing process at 350 °C for all annealing times. At this temperature, the translocation of the neutral axis Δ*h* was 0.31, 0.38, and 0.22 mm for *t* = 40, 90, and 150 min., respectively. For temperatures of 450 and 500 °C, the Δ*h* values were significantly reduced below the values recorded at a temperature of 250 °C. Annealing at 350 °C resulted in significant plasticization of the metal with a relatively small diffusion zone, which increased with temperature and weakened the strength of the bimetal.

Figure 9b shows the change in the value of the Δ*h* parameter with increasing annealing time. It shows that for a temperature of 350 °C, the annealing time *t* = 90 min. had the greatest impact, and its increase from 90 to 150 min contributed to a decrease in the Δ*h* value. The trends are opposite only for samples expressed at 500 °C. The annealing time at 250 °C had a slight impact on the Δ*h* values.

The weakest element of the Al/Cu bimetallic joint is, of course, the interface, especially when thermal interactions occur. During bending, the neutral axis moves away from the interface line towards the centre of gravity in the cross-section of the stronger Cu layer. The translocation of the interface to a greater distance from the neutral axis increases its load with increasingly higher normal stresses, destroying the bimetallic joint. This effect is particularly visible in SEM photographs, where brittle intermetallic segments are pushed out as a result of interface stretching. In addition to the normal stresses, the shear stresses occurring in this three-point bending pattern add to the normal stresses.

### 4.3. The Influence of the Annealing Temperature and Time on the Beginning of the Plasticization Process of the Extreme Al/Cu Layers

Permanent deformation occurring in the extreme (outer) layers of a three-point bent sample rarely reveals its origins in the bending curve in the force–deflection system (*F* − *d*). Typically, such a transition occurs gently (smooth curve), which does not provide grounds for determining the limit value of the moment that plasticizes the outer layers of the beam. In such a case, the conventional value of the moment corresponding to plastic deformation should be taken as *ε* = 0.2%. However, due to the simultaneous occurrence of elastic and plastic strains that generate the initial stresses, it is recommended to search for the value of the actual strain [37], which is defined as follows:(5)$$\varepsilon_{th}=\varepsilon_{0.2\%}\left[1-\frac{\varepsilon_{pr}\left(\varepsilon_{pr}+0.5\times \varepsilon_{0.2\%}\;\right)}{{\left(\varepsilon_{pr}+\varepsilon_{0.2\%}\;\right)}^{2}}\right],$$
where *ε*_0.2%_ is the plastic strain corresponding to the normative unit elongation (shortening) of the extreme bimetal layers equal to 0.002, and *ε*_pr_ is the maximum elastic strain corresponding to the proportional relationship between the loading force *F* and the maximum deflection *d* at the point of application of the load.

Charts of *F* − *d* obtained from three-point bending tests of the tested sample to determine the value of *ε*_pr_ were used, from which the value of force *F*_pr_ and the corresponding bending moment *M*_pr_ were determined from the following relationship:*M*_pr_ = 0.25 *F*_pr_ *l*,(6)
where *l* is the distance between supports, and *F*_pr_ is the force corresponding to deformation *ε*_pr_.

The value of the normal stress *σ*_pr_ (stress corresponding to the strain *ε*_pr_) in the extreme Al and Cu layers was calculated using the equation
*σ*_pr_ = *M*_pr_ × *h*_Al(Cu)_/*I*_w_,(7)
where *h*_Al(Cu)_ denotes the distances of the extreme aluminium and copper layers from the neutral axis.

Applying Hooke’s law to the stretched (compressed) extreme layers, the values of elastic strain *ε*_pr_ were calculated, and thus, the value of the actual strain *ε*_th_ was obtained from Equation (5). Next, for the deformation *ε*_th_, the corresponding deflection value *d*_th_ was determined from the relationship *d*_th_ = *ε*_th_ *l*^2^/12 *h*_Al(Cu)_. Further, using the *F* − *d* diagram (eliminating elastic deformations), the value of the force *F*_p_ was determined, which is the actual initiator of the onset of plasticity in the extreme bimetal layer. In turn, using the analogous equation to (6), the values of the actual bending moment *M*_p_ initiating plastic deformations in the most loaded layers (compressed) were determined. The influence of the annealing temperature and time on the variability in the plasticizing moment of the outer layers of the Al/Cu bimetal is shown in Figure 10, and intermediate values for determining *M*_p_ are included in Table 5.

At a temperature of 250 °C, the values of the plasticizing moment *M*_p_ for annealing times of 40, 90, and 150 min. were similar and amounted to over 12 Nm (Figure 10a and Table 5). They are slightly smaller than the value of *M*_p_ = 13.6 Nm for unannealed samples (*T* = 22 °C). At an annealing temperature of 350 °C, a two-fold decrease in the *M*_p_ value was observed compared to that at a temperature of 250 °C, and increasing the annealing time slightly decreased the torque value. For temperatures of 450 °C and 500 °C, the torque *M*_p_ was slightly higher than 4 Nm. In the case of a temperature of 500 °C and a time of *t* = 150 min., the moment *M*_p_ slightly increased to a value of 4.6 Nm.

The annealing time slightly influenced the *M*_p_ values at the tested temperatures (Figure 10b). The exception here is the temperature of 350 °C, for which the value of the moment *M*_p_ was 5.6 Nm for *t* = 40 min. and then dropped to 4.5 Nm and remained almost unchanged. 

To sum up the above, an annealing process at a temperature of 350 °C and above causes a significant reduction in the elastoplastic properties of the Al/Cu bimetal, which was demonstrated by the three-point bending test.

The springback of the Al/Cu bimetal is largely influenced by the difference in strength (including modulus of elasticity) between the component layers, which, in the case studied, increased with the annealing temperature. Additionally, the relative positions of the layers in the bending process are important. In the case studied, the hard copper layer was essentially compressed, and the ductile aluminium layer was stretched. The thickness ratio was irrelevant in the case studied because the thicknesses of the two layers were similar. The increase in the springback angle with temperature was, therefore, dependent on the relationship between the angles in the elastoplastic and plastic states, as shown in Table 3.

The technological process of joining copper and aluminium into a bimetal by rolling required significant loads, which resulted in large deformations and the increased strength of the Al/Cu composite, including improved elastic–plastic properties. The bimetal components, i.e., aluminium and copper, are characterized by a relatively low annealing temperature, which causes their rapid softening and increased ductility. The use of the annealing process at a temperature of 350 °C and higher in relation to the Al/Cu bimetal in the experiment caused the above phenomena, which resulted in increased flexibility, lower yield strength, and a general decrease in strength properties. The effect of this is a significant decrease in the value of the yield moment *M*_p_ of the outer layer of the samples in the three-point bending test. In the case of a temperature of 350 °C and an annealing time *t* = 40 min, it was 59% compared to the unannealed bimetal.

## 5. Microscopic Observations of Changes at the Interface of Al/Cu Layers 

Microscopic studies included observing the structure and measuring the thickness of the diffusion layer formed as a result of thermal interactions at the junction of the aluminium and copper layers. They were carried out on the side surface of the sample, where the highest bending moment occurred. Observations were made using a Phenom XL electron microscope. The thicknesses of the diffusion layers were measured by a graphical analysis of microscopic photographs using the Image Pro program. Additionally, the obtained data were verified by a linear analysis of the chemical composition in the diffusion zone. The measurement results of the thickness of the diffusion zone depending on the temperature and annealing time are illustrated in Figure 11a,b.

Microscopic observations showed that, in the case of unheated samples (*T* = 22 °C) and samples heated at 250 °C for 40, 90, and 150 min, there was no diffusion phenomenon at the border between Al and Cu, or its level was unregistrable. It can, therefore, be assumed that the above temperature was not exceeded during the rolling and joining of Al/Cu components by the sheet metal manufacturer, and it did not cause diffusion in the joint.

With the increase in the annealing temperature in the range of 350–500 °C, the thickness of the diffusion layer increased exponentially from zero to maximum values of 15.4 mm, 26.8 mm, and 31.5 mm for times *t* = 40, 90, and 150 min., respectively (Figure 11a). Similar shapes of the curves describing the growth of diffusive intermetallic layers with increasing temperature and annealing time were noted by Lee et al. [38], who identified two diffusive phases, AlCu and CuAl_2_, at the interface. In the observed microstructure, three intermetallic layers (intermetallic compounds—IMCs) with different shades can be clearly distinguished. Similar observations can also be found in the work of Wang et al. [39], who specified the third phase of the Cu_9_Al_4_ compound. In the case of a temperature of 500 °C, the increase in the thickness of the diffusion zone is two-fold from 40 to 150 min. Figure 11b shows that for a temperature of 350 °C, there is no increase in intermetallic phases with an increase in time *t.* For a temperature of 450 °C, there is an increase of 50% in the thickness of IMCs with an increase in time from *t* = 90 to 150 min.

The diffusion of aluminium into copper produced three clearly visible diffusion zones differing in shade, as illustrated by SEM photographs at ×2500 magnification, shown in Figure 12, Figure 13 and Figure 14. Observation of the diffusion zone at the point of maximum bending (Figure 12a) created after annealing at 350 °C for 40 min. allows us to conclude that the first newly formed thin IMC layer is brittle, and high tensile stress causes its degradation into small elements. Increasing the annealing time at this temperature causes a slight increase in the thickness of the diffusion zone, which cracks transversely when mechanically loaded (Figure 12b) for *t* = 90 min. and across and along for *t* = 150 min. (Figure 12c). For samples annealed at 450 and 500 °C, the neutral axis in the cross-section shifts to its initial position, which results in a reduction in load (stress), and the structure of the diffusion layer is more cohesive (Figure 13b,c and Figure 14). In the SEM photographs (Figure 13c and Figure 14b), one can see the local effects of mechanical loading with a highly fragmented diffusion layer.

SEM analysis of the photographs shown in Figure 12, Figure 13 and Figure 14 showed that at a temperature of 350 °C and higher, a thin and brittle diffusion layer was formed in the Al/Cu bimetallic samples at the interface. The graph shown in Figure 9 informs us that at this temperature, the greatest translocation of the neutral axis (in the most loaded cross-section of the sample) occurred in relation to the unloaded state. This effect occurred for all heating times: *t* = 40, 90, and 150 min. The displacement of the axis took place towards the centre of gravity in the cross-section of the copper layer. In this way, the diffusion zone is loaded with increased normal stress, and shear stress occurs at this level at the same time. As a result of the occurrence of increased equivalent stress, the continuity of the diffusion zone was destroyed and fragmented, which is indicated by SEM photographs in Figure 12. At higher annealing temperatures, i.e., 450 °C and 500 °C, there was a significant increase in the width of the diffusion zone, and at the same time, the neutral axis returned to the initial position. The diffusion zone in Figure 13 and Figure 14 is more continuous, and its cracks are only local.

## 6. Conclusions

Based on the above, general conclusions were formulated:The annealing of Al/Cu bimetal elements at a temperature of 250 °C for 40–150 min and subsequent cooling to room temperature does not cause significant changes in either the structure of the composite or the basic values of its mechanical parameters. There is, of course, at this temperature and for *t* = 150 min., a slight decrease of 3 GPa in the value of Young’s modulus, as well as a decrease of 1.3 Nm in the moment *M*_p_ plasticizing the extreme layers of Al/Cu components to the level of permanent deformation of 0.2%, a decrease of 0.06 mm in the value of the translocation parameter Δ*h* of the neutral axis, and an increase of approximately 0.17 deg. in the springback angle Δ*Θ*, on average.Significant changes in the values of the tested mechanical parameters of Al/Cu in relation to unheated samples occurred at the temperature *T* = 350 °C. The brittle diffusion zone then appeared, increasing with the annealing time to a level of 5.7 mm. At this temperature and for *t* = 150 min., the value of Young’s modulus decreased by 10.7 GPa, and the value of the moment *M*_p_ decreased by three times. Al/Cu samples annealed at 350 °C and for *t* = 90 min. were characterized by an increased neutral axis translocation parameter Δ*h*, which amounted to 0.38 mm. There was also a sevenfold increase in the spring angle, reaching a value of 2.8 deg.Annealing at temperatures of 450 and 500 °C resulted in an exponential increase in the thickness of the diffusion zone to 31.5 mm compared to a temperature of 350 °C, with noticeable increases in the tested mechanical parameters and even decreases in some values, i.e., Δ*Θ* and Δ*h.*

## Figures and Tables

**Figure 1 materials-17-04637-f001:**
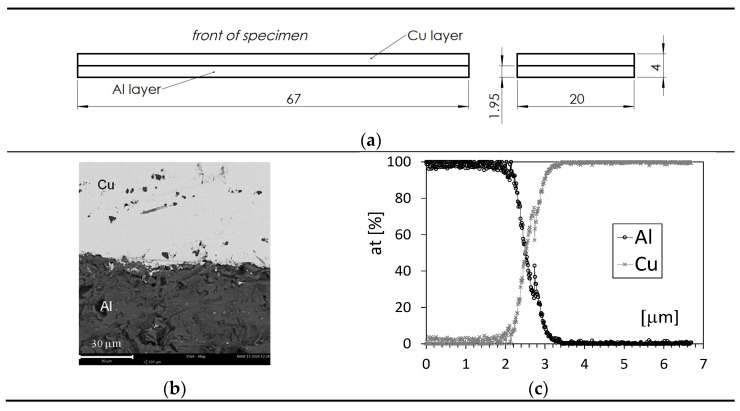
Characteristics of the test sample: (**a**) geometric dimensions; (**b**) an SEM photograph of the junction zone of copper and aluminium layers before heating; (**c**) the linear distribution of aluminium and copper at the interface before heating.

**Figure 2 materials-17-04637-f002:**
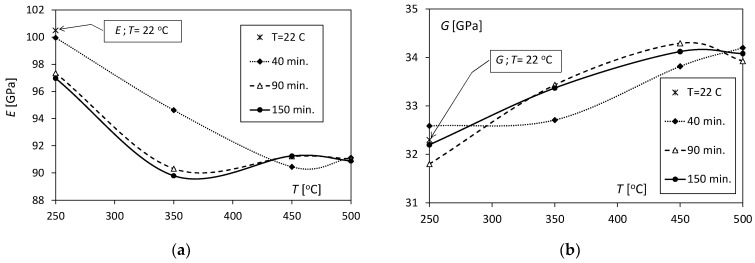
Changes in the values of the elasticity coefficients of the Al/Cu bimetal with temperature for various annealing times: (**a**) Young’s moduli; (**b**) shear moduli.

**Figure 3 materials-17-04637-f003:**
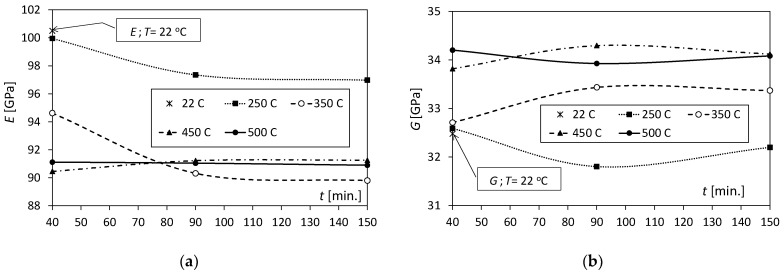
Dependence of the values of elasticity coefficients *E* and *G* on annealing times for various temperatures: (**a**) Young’s moduli; (**b**) shear moduli.

**Figure 4 materials-17-04637-f004:**
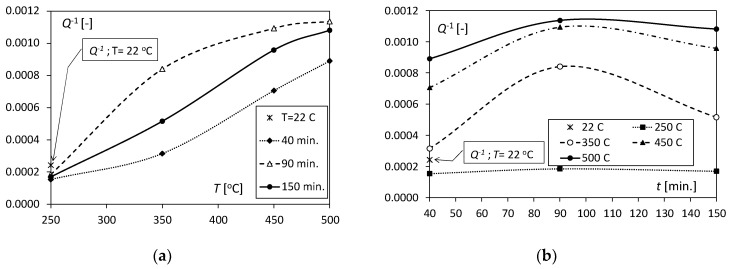
Values of the internal friction parameter *Q*^−1^ determined after thermal exposure (**a**) at different temperatures and (**b**) for different heating times.

**Figure 5 materials-17-04637-f005:**
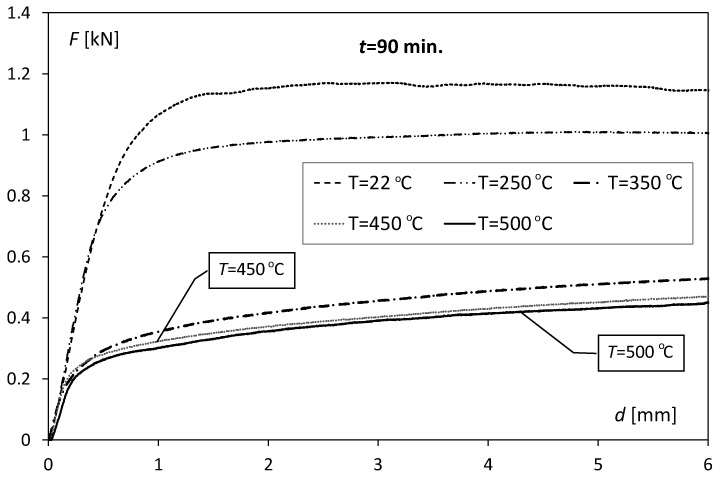
Examples of 3-point bending curves obtained for unheated samples (*T* = 22 °C) and samples heated for 90 min at temperatures *T* = 250, 350, 450, and 500 °C.

**Figure 6 materials-17-04637-f006:**
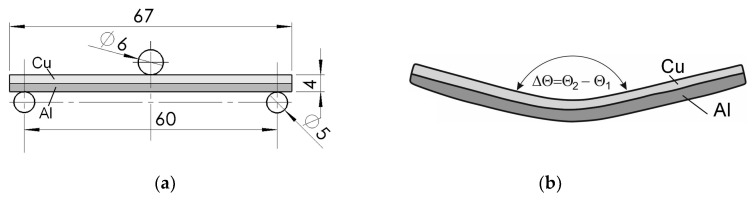
The scheme of 3-point bending of Al/Cu bimetallic samples: (**a**) the scheme of supporting and loading the Al/Cu sample; (**b**) the method of measuring the angle Δ*Θ*.

**Figure 7 materials-17-04637-f007:**
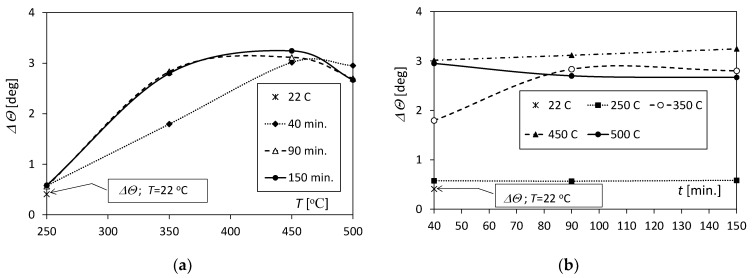
Springback of Al/Cu samples due to unloading: (**a**) influence of temperature increase; (**b**) influence of annealing time.

**Figure 8 materials-17-04637-f008:**
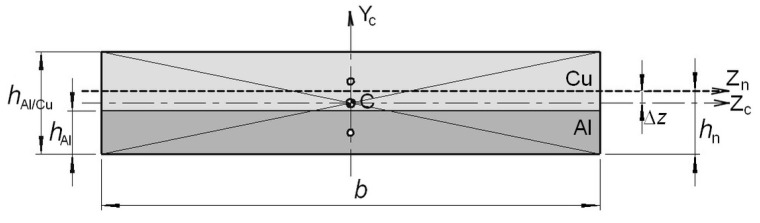
Designations of geometric dimensions of the Al/Cu bimetallic cross-section.

**Figure 9 materials-17-04637-f009:**
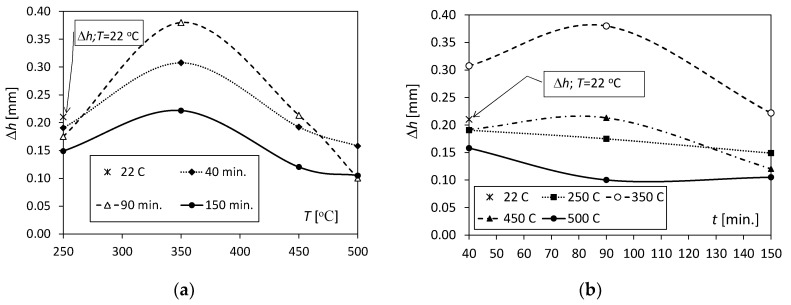
The change in the position of the neutral axis Δ*h* due to (**a**) temperature increase and (**b**) annealing time.

**Figure 10 materials-17-04637-f010:**
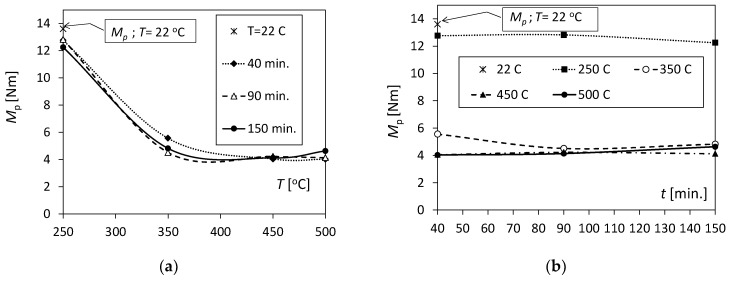
Variation in the bending moment *M*_p_ under the influence of (**a**) temperature and (**b**) annealing time.

**Figure 11 materials-17-04637-f011:**
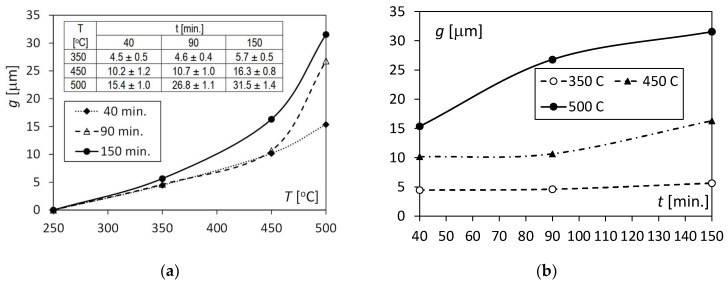
Changes in the thickness *g* of the diffusion zone: (**a**) under the influence of temperature; (**b**) under the influence of annealing time.

**Figure 12 materials-17-04637-f012:**
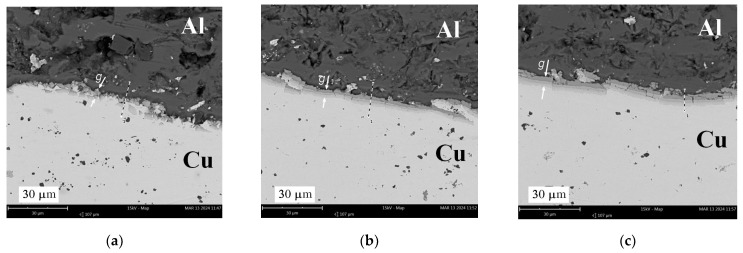
SEM photographs (×2.5k) of the connection of Al and Cu layers after heating at 350 °C: (**a**) 40 min.; (**b**) 90 min.; (**c**) 150 min.

**Figure 13 materials-17-04637-f013:**
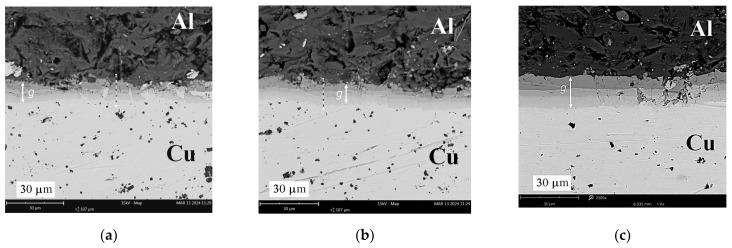
SEM photographs (×2.5k) of the connection of Al and Cu layers after heating at 450 °C: (**a**) 40 min.; (**b**) 90 min.; (**c**) 150 min.

**Figure 14 materials-17-04637-f014:**
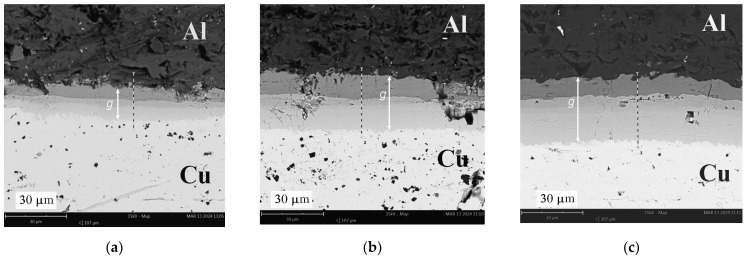
SEM photographs (×2.5k) of the connection of Al and Cu layers after annealing at 500 °C: (**a**) 40 min.; (**b**) 90 min.; (**c**) 150 min.

**Table 1 materials-17-04637-t001:** The chemical composition of the aluminium layer (Al).

Al	Fe	Si	Ti	Zn	Cu
%					
99.75	0.18	0.1	0.009	0.004	0.001
*Chemical composition according to Polish standard PN−77/H−82160.*					

**Table 2 materials-17-04637-t002:** The chemical composition of the copper layer (Cu).

Cu + Ag	Bi	Pb	Sb	As	Fe	Sn	Zn	Others
%								
99.98	0.001	0.003	0.001	0.001	0.002	0.002	0.001	0.009
*Chemical composition according to Polish standard PN−77/H−82120.*								

**Table 3 materials-17-04637-t003:** Average values of deflection angles *Θ*_1_ and *Θ*_2_ and their difference Δ*Θ* for the elastoplastic and plastic states measured in the tested temperature ranges and annealing times.

Temperature [°C]	Annealing Time *t* [min.]	Angle *Θ*_2_ [deg.] (Elastoplastic State)	Angle *Θ*_1_ [deg.] (Plastic State)	Angle Increment Δ*Θ* [deg.]
22	-	10.42	10.01	0.41
250	40	10.76	10.18	0.58
	90	10.75	10.18	0.57
	150	10.73	10.15	0.58
350	40	11.50	9.70	1.79
	90	12.48	9.65	2.83
	150	12.46	9.66	2.80
450	40	12.79	9.77	3.01
	90	12.81	9.69	3.11
	150	12.90	9.65	3.24
500	40	12.79	9.84	2.95
	90	12.44	9.74	2.70
	150	12.55	9.89	2.67

**Table 4 materials-17-04637-t004:** Quantities determined from the calculation of the translocation of the neutral axis of the Al/Cu bimetal for the tested loading conditions.

*T*	*t*	Δ*z*	*h_n_* _1_	*h_n_* _2_	Δ*h*
[°C]	[min.]	[Nm]	[mm]	[mm]	[mm]
22	-	0.45	2.47	2.68	0.21
250	40	0.45	2.46	2.65	0.19
	90	0.44	2.48	2.66	0.17
	150	0.45	2.45	2.60	0.15
350	40	0.42	2.44	2.74	0.31
	90	0.42	2.44	2.82	0.38
	150	0.40	2.39	2.61	0.22
450	40	0.47	2.47	2.67	0.19
	90	0.42	2.43	2.65	0.21
	150	0.45	2.44	2.56	0.12
500	40	0.45	2.45	2.60	0.16
	90	0.46	2.47	2.57	0.10
	150	0.45	2.46	2.57	0.11

**Table 5 materials-17-04637-t005:** Average values of various mechanical quantities determined for Al/Cu bimetal.

*T*	*t*	*F* _pr_	*M* _pr_	*ε* _pr_	*ε* _th_	*d* _pr_	*d* _th_	*M* _p_
[°C]	[min.]	[kN]	[Nm]	[mm/mm]	[mm/mm]	[mm]	[mm]	[Nm]
22	-	0.56	8.37	$2.13\times 10^{-3}$	$1.22\times 10^{-3}$	0.233	0.382	13.6
250	40	0.48	7.25	$1.85\times 10^{-3}$	$1.29\times 10^{-3}$	0.250	0.388	12.8
	90	0.41	6.18	$1.56\times 10^{-3}$	$1.37\times 10^{-3}$	0.257	0.376	12.8
	150	0.45	6.71	$1.77\times 10^{-3}$	$1.31\times 10^{-3}$	0.254	0.387	12.3
350	40	0.12	1.76	$4.67\times 10^{-4}$	$1.77\times 10^{-3}$	0.333	0.375	5.6
	90	0.11	1.67	$4.63\times 10^{-4}$	$1.78\times 10^{-3}$	0.334	0.376	4.5
	150	0.12	1.86	$5.37\times 10^{-4}$	$1.74\times 10^{-3}$	0.330	0.378	4.8
450	40	0.13	1.94	$5.40\times 10^{-4}$	$1.74\times 10^{-3}$	0.339	0.389	4.1
	90	0.14	2.10	$5.81\times 10^{-4}$	$1.72\times 10^{-3}$	0.324	0.376	4.2
	150	0.12	1.82	$5.10\times 10^{-4}$	$1.76\times 10^{-3}$	0.344	0.392	4.1
500	40	0.12	1.77	$4.99\times 10^{-4}$	$1.76\times 10^{-3}$	0.342	0.388	4.0
	90	0.16	2.33	$6.49\times 10^{-4}$	$1.69\times 10^{-3}$	0.329	0.389	4.1
	150	0.14	2.13	$5.79\times 10^{-4}$	$1.73\times 10^{-3}$	0.332	0.385	4.6

## Data Availability

No new data were created or analyzed in this study. Data sharing is not applicable to this article.

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
