# Peer review of "Thermal Annealing Effect on Elastoplastic Behaviour of Al/Cu Bimetal during Three-Point Bending"

_materials, 2024, doi:10.3390/ma17184637_

Round 1
Reviewer 1 Report
Comments and Suggestions for Authors
The effect of annealing on the elastoplastic properties of Al/Cu bimetal during three-point bending was investigated in this study. This work has clear logic and accurate language. It can be accepted after being revised according to the comments below:
1. Although the thickness of the Al layer is introduced in Figure 1a, it is recommended to state it in the text as well.
2. What is the reason for the change in elastic modulus and shear modulus after 450 ℃? Can you provide a detailed description and explanation in the study?
3. There are problems with the temperature unit in Figure 5. Please correct them.
4. Different measurement locations will result in different thicknesses of the diffusion zone. Please add error bars in Figure 11.
5. The overall results and discussion mainly describe the results and phenomenon (the changes of Young's moduli, shear moduli, and spring back etc.), but the discussion of the mechanism or mechanism that produces this phenomenon is not in-depth enough. Please discuss it in more depth.
6. The connection between changes in macroscopic mechanical properties and changes in diffusion zone should be described more clearly.
Comments on the Quality of English LanguageMinor editing of English language required.
Author Response
I want to thank you very much for your insightful review.
Changes suggested by the reviewer are marked in green in the manuscript text.
- Although the thickness of the Al layer is introduced in Figure 1a, it is recommended to state it in the text as well.
I would like to inform you that I have marked the nominal value of the Al layer - line numbers 98-99 in the text.
- What is the reason for the change in elastic modulus and shear modulus after 450 ℃? Can you provide a detailed description and explanation in the study?
Annealing at a temperature of 450 oC and above causes small changes in Young's modulus E (Fig. 2a), with values in the range of 90.5-91.3 GPa. This is the level of measurement uncertainty. A similar situation occurs in the case of the shear modulus G, whose values range from 33.8-34.3 GPa (Fig. 2 b). For aluminium, the temperature of 450 oC is very high and constitutes 70% of the melting temperature. In these conditions, the Al layer is thermally weakened, and the slightly thicker copper layer mainly determines the elastic properties of the Al/Cu bimetal. Above 450 oC, the effect of strain hardening resulting from the technological process of joining Al and Cu is substantially reduced.
- There are problems with the temperature unit in Figure 5. Please correct them.
I corrected the temperature units in degrees Celsius in Figure 5. I also changed the appearance of the lines recorded from the 3-point bending test. Additionally, I marked the curves that are close to each other with an appropriate annotation.
- Different measurement locations will result in different thicknesses of the diffusion zone. Please add error bars in Figure 11.
I share the reviewer's opinion that the thickness of the diffusion zone is different in different places of the interface. The measurement of the diffusion zone thickness was carried out by digital image analysis using the Image Pro Plus 6.0 program. In the analysed image zone, 20 points were selected for which the Al/Cu interface thickness was measured. I had difficulty in plotting error bars on the curves showing the variation of the diffusion zone thickness (Figure 11) in MS Excel. To remedy this situation, I have included in Figure 11 a table of average values of the diffusion zone thickness (g) along with the measurement uncertainty.
- The overall results and discussion mainly describe the results and phenomenon (the changes of Young's moduli, shear moduli, and spring back etc.), but the discussion of the mechanism or mechanism that produces this phenomenon is not in-depth enough. Please discuss it in more depth.
An attempt at a detailed description of the phenomenon related to changes of elasticity modules, the spring back effect and plasticization bimetal Al/Cu with consideration of mechanisms are included in the text of the manuscript (green colour). Line:439-461
- The connection between changes in macroscopic mechanical properties and changes in diffusion zone should be described more clearly.
In the manuscript, I included text describing the dependencies resulting from the change in thermomechanical properties of the Al/Cu bimetal that are related to the translocation of the neutral axis and the influence on the diffusion zone. (green colour), Line 512-823

Reviewer 2 Report
Comments and Suggestions for Authors
It is meaningful to investigate the effect of temperature and time of annealing on the elastoplastic properties of bimetallic aluminium-copper sheets. It could be accepted for publication, however, several details in the article are of concern.
1. Figure 1(c) looks unclear, please kindly replace a clear picture.
2. In Figure 5, the units “oC” is wrong, and the curves of 450 oC and 500 oC are not well distinguished.
3. In Figure 12, the diffusion is not obvious, how do you determine the thickness of the diffusion layer? Is it the length of the dotted line?

Author Response
I want to thank you very much for your review.
- Figure 1(c) looks unclear, please kindly replace a clear picture.
I have revised Figure 1c for better readability.
- In Figure 5, the units “oC” is wrong, and the curves of 450 oC and 500 oC are not well distinguished
I corrected the temperature units in degrees Celsius in Figure 5. I also changed the appearance of the lines recorded from the 3-point bending test. Additionally, I marked the curves that are close to each other with an annotation.
- In Figure 12, the diffusion is not obvious, how do you determine the thickness of the diffusion layer? Is it the length of the dotted line?
The measurement of the diffusion zone thickness was carried out by digital image analysis using the Image Pro Plus 6.0 program. In the analysed image zone, 20 points were selected for which the Al/Cu interface thickness was measured. In Fig. 12, the diffusion zone is small but recordable and amounted to g=4.5 micrometres for the temperature of 350 oC and t=40 min. In order to document the increase of the diffusion zone with temperature, the same SEM magnification of the photograph was assumed.
The dashed line in Figure 12 (also in Figures 13 and 14) indicates the direction of linear analysis of chemical composition. These drawings are only sample photographs of the side surfaces of the tested samples corresponding to different temperatures and annealing times. In Figures 12, 13, 14 I have added markers illustrating the method of measuring the diffusion layer thickness.

Reviewer 3 Report
Comments and Suggestions for Authors
This article presents a study on the elastoplastic deformation behavior of aluminum/copper bimetallic sheets under mechanical and thermomechanical loading conditions. The authors provide experimental results on the elastoplastic deformation behavior accompanied by a relevant theoretical background. The study investigates the springback effect, neutral axis displacement, and extreme layer plasticization moment values under various annealing conditions.
The manuscript has strengths in terms of the research motivation and the validity of the research methods. The study provides valuable information that can be practically applied in related industries. However, there are several errors in the presentation and analysis of the experimental results, which require significant revisions.
The following are the related review comments:
- In Tables 1 and 2, the standard for the chemical composition (%) should be clearly presented.
- In Figure 1 (a), it should be noted whether each rectangular solid is a side or top view.
- On page 3, the sentence "Unheated samples were also tested (T = 20 ℃)" and the sentence "Some samples were tested without thermal effects" should be revised to have the same sample.
- On page 4, when describing Figure 2a, it is stated that the samples heat-treated for 90 and 150 minutes decreased in value from 97 GPa to 90 GPa in the temperature range of 250-350 ℃, but it is incorrectly written as 250-450 ℃, so revision is needed.
- Additionally, in describing the results of Figure 3a, Figure 7, and Figure 9, there are numerical errors, so revision is needed, and the description should be clear in terms of meaning.
- In Figure 5, the 450 ℃ and 500 ℃ curves are difficult to distinguish, so revision is needed.
- On page 13, the sentence "At a temperature of 250 ℃, the values of the plasticizing moment Mp for annealing times of 40, 90 and 150 min. (Figure 10 a) were similar and amounted to over 12 Nm (Figure 12a)." should be revised to match Figure 10a and Figure 10b, but it is incorrectly matched with Figure 12a, and there is a sentence error due to incorrect use of a period(.), so revision is needed.
- In addition to the mentioned points, it is recommended to ensure consistency in the description of experimental results throughout the paper and to have it edited by a professional English editing service, and to check for typos.
Author Response
I want to thank you very much for your insightful review.
Changes suggested by the reviewer are marked in yellow in the manuscript text.
- In Tables 1 and 2, the standard for the chemical composition (%) should be clearly presented.
I have additionally provided information in Tables 1 and 2 about the standard according to which the chemical composition of aluminium and copper was determined. The chemical composition of the copper layer in the Al/Cu bimetal was determined according to the Polish standard PN-77/H-82120, and the chemical composition of aluminium according to PN-77/H-82160. (yellow colour) line: 112,113
- In Figure 1 (a), it should be noted whether each rectangular solid is a side or top view.
I made a new drawing 1a and marked the view type (front view).
- On page 3, the sentence "Unheated samples were also tested (T = 20 ℃)" and the sentence "Some samples were tested without thermal effects" should be revised to have the same sample.
My mistake. I corrected the text to make it clearer.
„The first group consisted of samples that were tested after heating at temperatures of 250, 350, 450, and 500 oC for 40, 90, and 150 minutes. The second group consisted of samples that were not subjected to the heating process (marked in the paper as 22oC).” (yellow colour) line:108-111
- On page 4, when describing Figure 2a, it is stated that the samples heat-treated for 90 and 150 minutes decreased in value from 97 GPa to 90 GPa in the temperature range of 250-350 ℃, but it is incorrectly written as 250-450 ℃, so revision is needed.
I have corrected the text in the manuscript to be consistent with Figure 2a mentioned by the reviewer. (yellow colour) line:168-174
- Additionally, in describing the results of Figure 3a, Figure 7, and Figure 9, there are numerical errors, so revision is needed, and the description should be clear in terms of meaning.
I have corrected numerical errors in the descriptions of Figures 3 a, 7 and 9. (yellow colour) line:175-178, 280-283, 366-377
- In Figure 5, the 450 ℃ and 500 ℃ curves are difficult to distinguish, so revision is needed.
Figure 5 has been corrected. Additionally, the densely spaced 450 oC and 500 oC curves have been marked with appropriate annotations.
- On page 13, the sentence "At a temperature of 250 ℃, the values of the plasticizing moment Mp for annealing times of 40, 90 and 150 min. (Figure 10 a) were similar and amounted to over 12 Nm (Figure 12a)." should be revised to match Figure 10a and Figure 10b, but it is incorrectly matched with Figure 12a, and there is a sentence error due to incorrect use of a period(.), so revision is needed.
I made a mistake. The reference to Figure 12a has no justification here, and I removed it. (yellow colour) line:425-428
- In addition to the mentioned points, it is recommended to ensure consistency in the description of experimental results throughout the paper and to have it edited by a professional English editing service, and to check for typos.
The above manuscript text was re-checked by an English lecturer at the university. I also checked the manuscript with Grammarly Premium and Open Writefull.
